# The Level of Intima-Media Thickness in Patients with Metabolic Syndrome in Poland Depending on the Prevalence of Type 2 Diabetes

**DOI:** 10.3390/biomedicines11061510

**Published:** 2023-05-23

**Authors:** Marcin Gierach, Roman Junik

**Affiliations:** 1Department of Endocrinology and Diabetology, Collegium Medicum in Bydgoszcz, Nicolaus Copernicus University in Toruń, ul. M. Skłodowskiej-Curie 9, 85-094 Bydgoszcz, Poland; 2Cardiometabolic Center Gierach-Med, ul. Bydgoskich Olimpijczyków 5/39-40, 85-796 Bydgoszcz, Poland

**Keywords:** MetS, DMt2, IMT, atherosclerosis, cardiovascular risk

## Abstract

Background: Metabolic syndrome (MetS), increasingly diagnosed among the Polish population, is a combination of factors that are associated with an increased risk of atherosclerosis and cardiovascular diseases. Intima-media thickness (IMT) of the common carotid artery has been suggested as, simply, a non-invasive and reproducible marker of the early stages of the atherosclerotic process. The carotid IMT can also be a strong predictor of future cerebral and cardiovascular events. The aim of our study was to evaluate atherosclerotic lesions in carotid vessels in patients with MetS depending on the presence of DMt2 and to assess which demographic factors affect the level of IMT. The study involved 335 subjects diagnosed with MetS, including 211 females (65%) and 124 males (37%) aged 37–82. The diagnosis of MetS was made on the basis of the International Diabetes Federation (IDF) criteria. The patients were divided into two subgroups: with DMt2 and without DMt2. The value of IMT depended on gender, education, and smoking status. We noticed that patients with DMt2 had the highest measurement of IMT compared with other groups (1.01 vs. 0.98). Additionally, a statistically significant difference between the subgroup with DMt2 and those without DMt2 was found (1.01 vs. 0.92; *p* < 0.005). Ultrasound assessment of the carotid IMT should be used more often in the diagnosis and monitoring of high cardiovascular risk and early progression of atherosclerosis, especially in patients with MetS with current DMt2.

## 1. Introduction

Metabolic syndrome (MetS) is a set of factors that increase the risk of developing atherosclerosis and vascular complications [1,2]. According to epidemiological data, there are approximately 10 million Poles in Poland who have confirmed MetS. The number of patients increases and, consequently, more people in our population may have complications of cardiovascular diseases.

Cardiovascular diseases (CVD) are the leading cause of premature death in developed countries. It is important to identify the patients at increased risk of these diseases. It is important to diagnose atherosclerotic complications at an early stage of development. One of the methods for this purpose is the evaluation of carotid arteries with the use of ultrasonography. To detect and monitor structural atherosclerotic lesions necessitates the measurement of the thickness of the middle and inner layers of the arteries, referred to as the intima-media thickness (IMT). This is a non-invasive and easy method for the estimation of arterial wall changes in early atherosclerosis. The measurement of the IMT of the common carotid artery by high-resolution ultrasound is assessed clinically. A value not exceeding 0.9 mm is considered a valid IMT. The thickness of the arterial wall defined in this way is treated as a determinant of the initial stage of the atherosclerotic process [3,4]. IMT is regarded as a marker of subclinical atherosclerosis and can be used as a predictor of cardiovascular risk [5].

Patients with MetS belong to the group of high or very high cardiovascular risk. It has been reported that MetS is associated with an alteration in the arterial system [6,7,8] and with inflammation [9,10,11]. In the general population, we have previously described that MetS accelerates arterial aging [12,13]. In the meta-analysis of 21 studies, Cuspidi et al. also claimed that MetS is a risk factor for early carotid atherosclerosis in members of the general population, regardless of sex [14]. However, among patients with MetS, there is a large variation in cardiovascular risk due to the presence or absence of its individual components, which impacts differentially on arterial abnormalities Many studies reveal that MetS components were associated with a two-fold higher odds ratio of presenting extremely stiff arteries, after controlling for age, sex, and the occurrence of diabetes mellitus type 2 (DMt2) [13,15]. Additionally, Jarvisalo et al. in their research showed that patients with DMt2 have a higher level of IMT [16]. It is connected with insulin resistance (IR) and endothelial dysfunction (ED), which is a well-known important risk factor for the development of diabetes cardiovascular complications [17,18]. The relationship between the severity of atherosclerotic lesions and the level of glycemic control has been demonstrated by Larsen’s team performing an 18-year follow-up of people with diabetes and an assessment of glycated hemoglobin [19]. Hyperglycemia affects many biochemical and cellular processes, causing intensification of oxidative stress and changes in lipid metabolism. These processes lead to the progression of atherosclerotic lesions [4]. The carotid IMT increases with the duration of DMt2 [20]. There are studies that show that carotid atherosclerosis parameters predict adverse cardiovascular outcome and improve risk stratification in DMt2 patients [21,22].

The aim of our study was to evaluate atherosclerotic lesions in patients with MetS depending on the presence of DMt2 and to assess which demographic factors affect the level of IMT.

## 2. Material and Methods

The study involved 335 subjects, including 211 females (63%) and 124 males (37%) aged 37–82 with recognized MetS. The diagnosis of MetS was made on the basis of the International Diabetes Federation (IDF) criteria (Table 1). The inclusion criteria were identification of 3 out of 5 components of MetS. All patients were recruited from the Department of Endocrinology and Diabetology Collegium Medicum University of Nicolaus Copernicus in Bydgoszcz, Poland, in a consecutive manner from 2020 to 2021. 

### 2.1. Procedures

MetS diagnosis was established when three or more IDF criteria were met. The characteristics of the study group are presented in Table 2. The patients were divided into two subgroups: with DMt2 (n = 191) and without DMt2 (n = 144). 

Anthropometric measurements including height, weight, and waist circumference (WC) were obtained from all participants [23]. Finally, the following demographic factors: age, sex, and obesity were determined. 

Systolic (SBP) and diastolic blood pressure (DBP) were measured in the sitting position after 15 min of rest using an appropriately sized cuff in both upper extremities [23]. In patients without previous diagnosis of hypertension, arterial hypertension was diagnosed according to the IDF definition (RR ≥ 130/85 mmHg). 

Venous blood samples were collected from fasting patients for biochemical analyses. Levels of fasting total plasma cholesterol (TC), triglycerides (TG), high-density lipoprotein cholesterol (HDL-C) and fasting blood glucose (FBG) were evaluated in all patients. Low-density-lipoprotein cholesterol (LDL-C) was calculated using the Friedewald formula. Non-high-density-lipoprotein cholesterol (non-HDL-C) was calculated on the basis of the formula: TC–HDL-C [23]. HbA1c concentration was also determined.

The diagnosis of DMt2 was made on the basis of 2 h PG value ≥ 200 mg/dL (11.1 mmol/L). Additionally, we diagnosed DMt2 on the basis of HbA1C ≥ 6.5% or in patients with classic symptoms of hyperglycemia or hyperglycemic crisis, a random plasma glucose ≥ 200 mg/dL (11.1 mmol/L) [15].

All tests were performed at the Department of Laboratory Medicine, Nicolaus Copernicus University, Collegium Medicum, Bydgoszcz, Poland, using a Horiba ABX Pentra 400 analyzer (Horiba ABX, Montpelier, France) [15].

Exclusion criteria: a history of heart surgery or other cardiovascular interventions, congenital defects of the heart, cardiac rhythm disorders, chronic kidney disease with GFR < 60 mL/min/1.73 m^2^, pregnancy, electrolyte disorders, inflammation, anemia, prostate disease, and Cushing’s syndrome [16]. Additionally, subjects after stroke and with dementia were excluded, as were those presenting other conditions that compromise cognition such as depression, anxiety, taking psychotropic drugs, psychiatric diseases, and history of alcohol or chemical addiction or uncorrected visual or hearing disorder [16].

### 2.2. Measurements of IMT

The measurement of the thickness of the middle and inner layers (intima-media thickness, IMT) of the carotid arteries was performed using an ultrasound scan with a 7 MHz Phillips-SD 800 linear probe by one radiologist. The measurements were assessed at the level of the common carotid artery at a height of 10 mm from the split of the common carotid artery to the external and internal carotid artery and in the areas free of atherosclerotic plaques. All measurements were performed twice and then averaged. A value not exceeding 0.9 mm is considered a valid IMT. An abnormal carotid IMT is generally defined as a thickness > 0.9 mm according to the 2013 European Society of Hypertension (ESH)/European Society of Cardiology (ESC) guidelines for hypertension for controlling cardiovascular risk factors in hypertensive patients [24]. Atherosclerotic plaques in the carotid artery were defined as a localized thickening of >1.2 mm that did not uniformly involve the entire left or right common carotid bifurcation with or without flow disturbance. Atherosclerotic changes in the carotid artery were defined as increased carotid IMT or presence of plaques [25].

### 2.3. Statistical Analysis 

Statistical analysis was carried out using the Statistica 8.0 software (Statsoft Poland, Bydgoszcz, Poland). For statistical evaluation, we used the Shapiro–Wilk test, the Student *t*-test, and the multiple linear regression model. The results were considered statistically significant for *p* < 0.005. 

### 2.4. Ethical Approval

All the procedures were performed in accordance with the 1964 Helsinki Declaration [26]. The research protocol was reviewed and approved by the Ethics Committee at the University Hospital in Bydgoszcz (Permission number KB/224/2022). All subjects granted their informed consent for participation in the study [26]. 

## 3. Results

The comparison of the subgroups is shown in Table 3.

In the study group of patients with MetS, DMt2 was present in 191 patients (57%). No statistical differences in age, waist circumference (WC), body mass index (BMI), systolic blood pressure (SBP), diastolic blood pressure (DBP), total cholesterol (TC), low-density lipoprotein cholesterol (LDL-C), and non-high-density lipoprotein cholesterol (non-HDL-C) between DMt2 and without DMt2 groups were found. The DMt2 subgroup, however, had a higher level of TG (1.91 vs. 1.77; *p* < 0.005) and a lower HDL-C level (1.02 vs. 1.14; *p* < 0.005).

We assessed the value of IMT according to gender, level of education, and smoking status. We noticed statistically significant differences in females, in a group of patients with primary education, and smoking status (Table 4).

We also divided the study group due to the age of the patients. The age of 65 was the cut-off point. Regardless of the age of the studied group of females, statistically significant differences were noticed (Figure 1 and Figure 2). 

The results of multiple linear regression, including IMT parameters and traditional cardiovascular risk factors, are shown in Table 5. The results of the analysis were presented separately for the subgroups with DMt2 and without DMt2. Age, SBP, and smoking status were independent predictors of mean IMT increase in both subgroups. Additionally, HDL-C and TG were independent predictors of mean IMT increase in the subgroup with DMt2.

## 4. Discussion

MetS is a set of factors that increase the risk of atherosclerosis and the development of CVD. It is important to diagnose atherosclerotic complications at an early stage of development, especially in patients at increased risk of these diseases. Complex IMT can be an important predictor of atherosclerosis and a higher rate is associated with an increased risk of heart attack and stroke [20,27,28].

According to the meta-analysis performed by Cusoidi et al. [14], the mean common carotid IMT was higher in MetS study participants compared with their non-MetS counterparts (759 ± 41 vs. 695 ± 27 μm), the standard mean difference being 0.39 ± 0.05 (confidence interval: 0.29–0.48, *p* < 0.0001). 

In our study, we have distinguished two subgroups of patients with MetS: with and without DMt2. We noticed a higher level of IMT in females in the first subgroup in comparison to the subgroup without DMt2. These differences were statistically significant (0.98 vs. 0.89; *p* < 0.005). In males, a higher level of IMT was also found in the subgroup with DMt2, but the differences were not statistically significant. This could be due to the smaller size of the study subgroups. Our findings are similar to the results obtained by Larsen et al. [19], performing an 18-year follow-up of people with diabetes and an assessment of glycated hemoglobin, which showed the relationship between the severity of atherosclerotic lesions and the level of glycemic control. IMT was significantly higher in diabetic patients than in an age- and sex-matched reference population. In addition, Jarvisalo in his research showed that patients with diabetes have a higher level of IMT [16]. 

In our study, we also observed that patients with DMt2 had a significantly higher level of TG and HbA1c and a lower level of HDL-C, which can explain the high IMT. Among patients with MetS, there is a large variation in cardiovascular risk due to the presence or absence of its individual components. Scuteri et al. [29] in their study evaluated that specific clusters of MetS components: high TG or low HDL-C, high blood pressure, abdominal obesity, or high glucose were accompanied by a 50–90% significantly greater likelihood of presenting extremely high intima-media thickness (via ultrasound of carotid artery). In other studies, a gradual increase was observed as the number of MetS components increased [30]. In addition, Kerimkulova et al. claimed that a greater number of MetS components, with abdominal obesity (AO) or without AO, is associated with a higher carotid IMT [31].

We also divided our study group, due to their age, into two subgroups. We noticed elevated carotid IMT values in the subgroup above 65 years. It was statistically significant only in females. Additionally, Lambrinoudaki et al. [32] in their study assessed the occurrence of changes in IMT in patients over 65 with MetS and without MetS. They found that IMT was higher in women with MetS (0.78 ± 0.12 mm vs. 0.74 ± 0.11; *p* = 0.003) and that it was already associated with subclinical atherosclerosis in the first postmenopausal decade.

Smoking habit is another independent risk factor for the pathogenesis and development of atherosclerosis. It is associated with the progression of carotid IMT, which may be related to the risk of atherosclerosis events in MetS patients [33]. However, the pathophysiological mechanisms of smoking status are confusing and complicated. In our study, we noticed that smokers with DMt2 had the highest index of IMT compared with other patients. Additionally, a statistically significant difference between the subgroup of smokers with DMt2 and those without DMt2 was found (1.01 vs. 0.92; *p* < 0.005). Of course, we know that cigarette smoking affects the development of atherosclerosis [34,35] and DMt2 intensifies this phenomenon but, in our study, we confirmed that this relationship also occurs in the group of patients at the highest cardiovascular risk.

The education level is another factor that we took into account. Our findings confirmed that in patients with primary education, statistically significant differences in the IMT index between the subgroup with DMt2 and without DMt2 were noted (0.99 vs. 0.93; *p* < 0.005). It could be partially connected with inadequate diet, uncontrolled DMt2, and irregular medication intake. Taking medications, including metformin, is crucial. Recent studies have reported that low socioeconomic status increases the risk of cardio-metabolic diseases, including DMt2, cardiovascular disease, and MetS [36]. Caturano et al. [37] in their study show that metformin in patients with DMt2 reduced the risk of all-cause mortality and age-related comorbidities such as cardiovascular disease. Our data also suggested that education may be considered an independent predictor of cardiovascular risk in MetS.

## 5. Limitation of our Study

The main limitation of the study is not considering pharmacological treatment due to the many medications taken by patients with MetS. Of course, metformin is the drug of first choice in DMt2 and prediabetes and most of the patients were taking it, but some were also taking insulin. The main drugs in hypertension were ACE inhibitors and, in lipid disorders, statins. The size of the study group was due to the specificity of our clinic but allowed for statistical research. We also did not consider the role of depression in IMT.

## 6. Conclusions

Ultrasound assessment of the carotid IMT should be used more often in the diagnosis and monitoring of high cardiovascular risk and early progression of atherosclerosis, especially in patients with metabolic syndrome with current type 2 diabetes. Carotid IMT value allows early detection of atherosclerotic lesions and enables the commencement of appropriate treatment, its monitoring, and allows the implementation of other preventive methods to improve the prognosis of patients with a high cardiovascular risk.

## Figures and Tables

**Figure 1 biomedicines-11-01510-f001:**
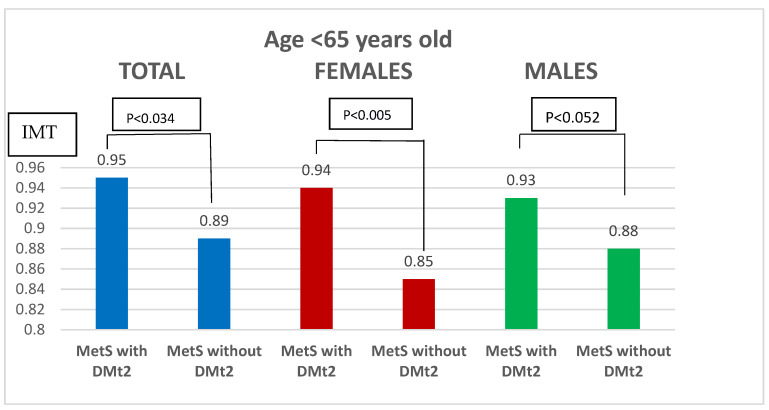
Comparison of IMT depending on the presence or absence of DMt2 in patients with MetS under the age of 65. IMT—intima-media thickness; MetS—metabolic syndrome; DMt2—diabetes mellitus type 2.

**Figure 2 biomedicines-11-01510-f002:**
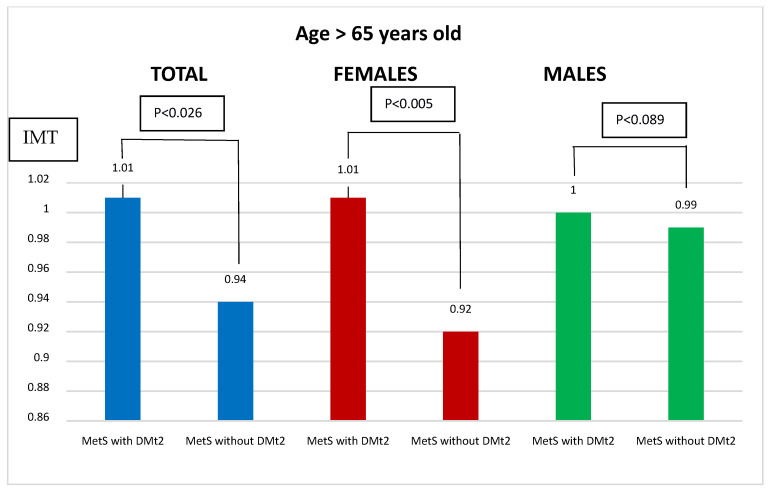
Comparison of IMT depending on the presence or absence of DMt2 in patients with MetS over the age of 65. IMT—intima-media thickness; MetS—metabolic syndrome; DMt2—diabetes mellitus type 2.

**Table 1 biomedicines-11-01510-t001:** IDF criteria of metabolic syndrome.

Abdominal obesity [cm]	F ≥ 80 or M ≥ 94
Arterial hypertension (HA) [mm Hg]	≥130/85 or treated for arterial hypertension
Triglycerides (TG) [mg/dL]	≥150 [1.7 mmol/L] or treated for dyslipidemia
HDL-C [mg/dL]	<50 [1.3 mmol/L] in women and <40 [1.0 mmol/L] in men
Fasting glycaemia (FG) [mg/dL]	≥100 [5.6 mmol/L] or treated for diabetes

**Table 2 biomedicines-11-01510-t002:** The characteristics of the study group.

Parameters		Total	Females	Males
N (%)		335	211 (63%)	124 (37%)
Age (y) ± SD		68.3 ± 4.7	71.7 ± 5.5	62.8 ± 3.5
WC (cm) ± SD		109 ± 6.3	107.1 ± 6.4	112.3 ± 6.1
BMI (kg/m^2^) ± SD		32.3 ± 3.1	31.9 ± 3.2	33.1 ± 2.9
SBP (mmHg) ± SD		133.5 ± 7.0	134.6 ± 6.7	131.9 ± 7.6
DBP (mmHg) ± SD		79.2 ± 3.5	78.9 ± 3.3	79.4 ± 3.8
HA (n%)		317 (94.6%)	203 (96.2%)	114 (92.0%)
TC (mmol/L) ± SD		5.44 ± 0.89	5.36 ± 0.84	5.58 ± 0.92
LDL-C (mmol/L) ± SD		3.52 ± 0.84	3.42 ± 0.82	3.68 ± 0.85
HDL-C (mmol/L) ± SD		1.07 ± 0.21	1.12 ± 0.23	0.93 ± 0.18
TG (mmol/L) ± SD		1.81 ± 0.67	1.76 ± 0.64	1.94 ± 0.73
Non-HDL-C (mmol/L) ± SD		4.37 ± 0.76	4.24 ± 0.72	4.65 ± 0.78
IFG (n; %)		81/335 (24%)	41/211 (19.4%)	40/124 (32.2%)
DMt2 (n; %)		191/335 (57.0%)	125/211 (59.3%)	66/124 (53.2%)
HbA1c (mmol/mol; %)	DMt2	7.8	7.6	7.9
	without	5.9	5.8	6.1
Education	Primary	86/335 (25.7%)	56/211 (26.5%)	30/124 (24.2%)
	Secondary	145/335 (43.3%)	86/211 (40.8%)	59/124 (47.6%)
	Higher	104/335 (31.0%)	64/211 (30.3%)	40/124 (32.26%)
Smoking status		75/335 (22.4%)	39/211 (18.5%)	36/124 (29.0%)

WC—waist circumference; BMI—body mass index; SBP—systolic blood pressure; DBP—diastolic blood pressure; HA—arterial hypertension; TC—total cholesterol; LDL-C—low-density lipoprotein cholesterol; HDL-C—high-density lipoprotein cholesterol; TG—triglycerides; non-HDL-C—non-high-density lipoprotein cholesterol; IFG—impaired fasting glucose; DMt2—diabetes mellitus type 2.

**Table 3 biomedicines-11-01510-t003:** Comparison of the study subgroups.

Parameters		DMt2	Without DMt2	*p* < 0.005
N (%)		191	144	*p* < 0.14
Age (y) ± SD		69.5 ± 5.7	67.7 ± 4.5	*p* < 0.23
WC (cm) ± SD		111 ± 6.9	106.1 ± 5.4	*p* < 0.047
BMI (kg/m^2^) ± SD		32.8 ± 2.9	31.3 ± 3.3	*p* < 0.036
SBP (mmHg) ± SD		136.2 ± 8.0	132.6 ± 6.2	*p* < 0.41
DBP (mmHg) ± SD		79.8 ± 3.6	78.3 ± 3.3	*p* < 0.35
HA (n%)		185 (96.8%)	132 (91.6%)	*p* < 0.008
TC (mmol/L) ± SD		5.47 ± 0.92	5.39 ± 0.84	*p* < 0.18
LDL-C (mmol/L) ± SD		3.55 ± 0.86	3.47 ± 0.82	*p* < 0.11
HDL-C (mmol/L) ± SD		1.02 ± 0.22	1.14 ± 0.25	*p* < 0.005
TG (mmol/L) ± SD		1.91 ± 0.77	1.77 ± 0.62	*p* < 0.005
Non-HDL-C (mmol/L) ± SD		4.39 ± 0.78	4.34 ± 0.72	*p* < 0.16
HbA1c (mmol/mol; %)	DMt2	7.8	5.9	*p* < 0.005
Education	Primary	45/191 (23.6%)	41/144 (28.5%)	*p* < 0.067
	Secondary	82/191 (42.9%)	63/144 (43.7%)	*p* < 0.043
	Higher	64/191 (33.5%)	40/144 (27.7%)	*p* < 0.086
Smoking status		39/191 (20.4%)	36/144 (25.0%)	*p* < 0.14

WC—waist circumference; BMI—body mass index; SBP—systolic blood pressure; DBP—diastolic blood pressure; HA—arterial hypertension; TC—total cholesterol; LDL-C—low-density lipoprotein cholesterol; HDL-C—high-density lipoprotein cholesterol; TG—triglycerides; non-HDL-C—non-high-density lipoprotein cholesterol; IFG—impaired fasting glucose; DMt2—diabetes mellitus type 2.

**Table 4 biomedicines-11-01510-t004:** Comparison of IMT depending on the presence or absence of DMt2 in patients with MetS.

IMT		n	MetS with DMt2	N	MetS without DMt2	*p* < 0.005
Males		66	0.97	58	0.93	*p* < 0.044
Females		125	0.98	86	0.89	*p* < 0.005
Total		191	0.98	144	0.91	*p* < 0.012
Education	Primary	62	0.99	24	0.93	*p* < 0.005
	Secondary	82	0.98	63	0.93	*p* < 0.038
	Higher	47	0.97	57	0.93	*p* < 0.043
Smoking status		29	1.01	46	0.92	*p* < 0.005

IMT—intima-media thickness; MetS—metabolic syndrome.

**Table 5 biomedicines-11-01510-t005:** The results of multivariate regression analysis of IMT in subgroups with DMt2 and without DMt2 with selected traditional risk factors.

Mean IMT				
R Model	0.244		0.283	
*p*	<0.005		<0.005	
	DMt2		Without DMt2	*p* < 0.005
Variable	Coeff.	*p*	Coeff.	*p*
Age	0.003	<0.005	0.004	<0.005
Sex	0.026	<0.16	0.031	<0.12
WC	0.008	<0.032	0.013	<0.045
BMI	0.012	<0.23	0.019	<0.56
SBP	0.002	<0.005	0.004	<0.005
DBP	0.032	<0.28	0.034	<0.52
HA	0.036	<0.22	0.038	<0.27
TC	0.014	<0.087	0.023	<0.13
LDL-C	0.006	<0.098	0.008	<0.066
HDL-C	0.002	<0.005	0.007	<0.009
TG	0.002	<0.005	0.012	<0.008
Non-HDL-C	0.007	<0.063	0.008	<0.059
Smoking status	0.004	<0.005	0.003	<0.005

WC—waist circumference; BMI—body mass index; SBP—systolic blood pressure; DBP—diastolic blood pressure; HA—arterial hypertension; TC—total cholesterol; LDL-C—low-density lipoprotein cholesterol; HDL-C—high-density lipoprotein cholesterol; TG—triglycerides; non-HDL-C—non-high-density lipoprotein cholesterol.

## Data Availability

Data is unavailable due to privacy or ethical restrictions.

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
