# Peer review of "The Level of Intima-Media Thickness in Patients with Metabolic Syndrome in Poland Depending on the Prevalence of Type 2 Diabetes"

_biomedicines, 2023, doi:10.3390/biomedicines11061510_

Round 1

Reviewer 1 Report

Gierach Marcin et al Metabolic evaluated the atherosclerotic lesions in carotid vessels in patients with metabolic syndrome (MetS) depending on the presence of DMt2 and to assess, which demographic factors affect the level of IMT. The study involved 335 subjects diagnosed with MetS, including 211 females (65%) and 124 males (37%) aged 37-82. Patients were divided into two subgroups: with DMt2 and without DMt2. The value of IMT was depending on gender, education and smoking status. Patients with DMt2 had the highest index of IMT compared with other groups (1.01 vs
0.98). Additionally, a statistically significant difference between the subgroup with DMt2 and those without DMt2 was found (1.01 vs 0.92; p<0.005).

The paper gvive some new information to literature data.

1) the authors should discuss the role of depression in IMT evaluation, if there are not this information please state it in study limitations (plase cite DOI: 10.1016/j.atherosclerosis.2014.01.012 and PMID: 23241128).

2) Did this patients are treated with SGLT2I?

Author Response

Dear Reviever,

Thank You very much for Your comments. We have corrected our study according to them.

  1. We corrected paragraph: study limitation.
  2. The patients were not treated with SGLT2 inhibitors.

Yours Faithfully

Marcin Gierach

Reviewer 2 Report

Thank you for the change to Review this paper. In this paper Gierach M investigated the relationship between intima media thickness, a non invasive ultraound parameter with a well known association with cardiovascular risk, and the presence/absence of type 2 diabetes. 

The topic is not original, and the paper reports only confirmatory results because it is well known the association between IMT and  Following some suggestion: Abstract
  • it makes not sense to put references in the abstract, please remove
  • Authors states IMTas an “index” but it is not the case; IMT is a misuration: please, remove index and add mm in all the paper text
  Introduction The language is very poor and the introduction sounds too much as a review of the literature. In my opinion should be cited studies investigating IMT in patients with early alteration of glucose homeostasis such as 10.1016/j.atherosclerosis.2012.05.008 and   Methods The most of this section is inappropriate. Ethical section is reported two times, remove the sentence in page 2. Table 2 and 3 should be placed in the results section, not here. Please, amend Furthermore, group division should be placed in result section Measurement of IMT should be placed in a separate paragraph. ESC guidelines: reference is missing   Results Authors should put all P values in tables; It is not acceptable just write NS. Figure1  is very confusing, please, choose a color for every category (total, female, male) and correct the figure. I noted that there were no available data on renal function of this patients. Renal function also in early stage has a good correlation with IMT ( 10.1016/j.numecd.2021.08.030) . Data on renal function (eGFR and/or presence/absence of proteinuria) should be reported if available. If not, this should be stated as a study limitation.   In my opinion it is not possible to ignore that current ESC guidelines do not recommend IMT measurement in clinical practice for the CVrisk evaluation in patients with type 2 diabetes and they recommend the utilization of the presence/absence of carodid plague. if the data is available it should be reported and commented, otherwise this is an important study limitation.

Author Response

Dear Reviever,

Thank You very much for Your comments. 

  1. We corrected our study as noted by Your recommendation:
    - we remove references in the abstract
    - we remove "index" and add "measurements"
    - we try to improve our language
    - we added some more information to the introduction - 10.1016j/j,atherosclerosis.2012.05.008
    - we remove ethical section
    - we placed table 3 in the results section
    - we placed in a separate section the measurement of IMT
    - we added ESC quidelines reference
    - we added p values in tables in results section
    - we corrected figures as noted by reviewers
    - we corrected section - study limitation

Yours Faithfully

Marcin Gierach

Round 2

Reviewer 1 Report

The manuscript has been adequately changed and all issues pointed out in my review have been satisfactorily addressed.

Author Response

Dear Reviewer

Thank You very much for Your all comments

Kinds Regard

Marcin Gierach

Reviewer 2 Report

No other comments

Author Response

(The authors gave the same response as above.)
